# Regulation of AP1 adaptor assembly by the bi-handed chaperone MEA1

Chun Wan[1], Jingyi Wu[1], Yan Ouyang[1], Harrison Puscher[1,4], Yuan Tian [2], Suzhao Li [3], Qian Yin [2] ✉ & Jingshi Shen [1] ✉

Bidirectional trafficking between the *trans*-Golgi network (TGN) and endolysosomal compartments lies at the intersection of biosynthetic and degradative pathways. At the center of this trafficking route is the adaptor protein complex 1 (AP1), a heterotetramer essential for cargo recognition and vesicle budding. Here, we identified Male-Enhanced Antigen 1 (MEA1), a previously uncharacterized protein, as a critical AP1 regulator. Loss of MEA1 resulted in depletion of AP1 subunits and impaired trafficking of AP1-dependent cargoes. Mechanistically, MEA1 acts as a bi-handed chaperone, simultaneously engaging and stabilizing the μ1 and β1 subunits of AP1. The MEA1-stabilized μ1 and β1 collide with the γ and σ1 subunits stabilized by Alpha- and Gamma-Adaptin Binding Protein (AAGAB), another bi-handed chaperone, leading to formation of the tetrameric AP1 adaptor and release of both chaperones. These findings identify MEA1 as a key AP1 regulator and uncover a dual chaperone collision mechanism potentially generalizable to multiprotein complex assembly.

Bidirectional trafficking between the *trans*-Golgi network (TGN) and endolysosomal compartments represents a unique branch of the eukaryotic endomembrane system, lying at the intersection between biosynthetic and degradative pathways[1–5]. At the center of this trafficking route is the adaptor protein complex 1 (AP1), a heterotetramer composed of two large subunits (γ and β1), one medium subunit (μ1), and one small subunit (σ1)[1,6,7]. AP1 is recruited from the cytosol to the TGN and endosomal membranes through coincident detection of specific membrane lipids and activated ADP-ribosylation factor (ARF) GTPases[7,8]. Once membrane-associated, AP1 recognizes sorting signals —such as tyrosine- or dileucine-based motifs—in the cytosolic domains of cargo proteins[9–14]. Upon cargo engagement, AP1 undergoes conformational changes that promote clathrin recruitment and initiate vesicle budding[6,15–20].

AP1, initially known for transporting constitutive cargoes such as lysosomal enzymes between the TGN and endolysosomal compartments, also regulates cellular signaling. In the cytosolic DNA-sensing pathway, cyclic GMP-AMP synthase (cGAS) produces cGAMP, which activates Stimulator of Interferon Genes (STING)[21–23]. Activated STING translocates from the endoplasmic reticulum to the Golgi, recruits TANK-binding kinase 1 (TBK1), and drives IRF3 phosphorylation to induce type I interferons and pro-inflammatory cytokines[21,22]. To prevent excessive activation, STING is removed from signaling compartments via AP1-mediated lysosomal degradation[23]. AP1 recognizes a dileucine motif in STING's cytosolic tail, with TBK1-dependent phosphorylation enhancing this interaction and promoting STING incorporation into clathrin-coated vesicles (CCVs) that deliver phosphorylated STING and TBK1 to lysosomes[23]. This positions AP1 as both a cargo adaptor and a checkpoint limiting innate immune signaling.

Although the core components of AP1 and their roles in membrane trafficking and signaling are well established, only recently have we begun to understand how the AP1 complex itself is assembled[24–30]. It is now evident that adaptor protein (AP) complexes do not assemble spontaneously in cells, as once presumed; instead, their biogenesis is a coordinated process directed by assembly chaperones. These chaperones are mechanistically distinct from folding chaperones such as Hsp70: rather than assisting with polypeptide folding, they bind

[1]Department of Molecular, Cellular and Developmental Biology, University of Colorado, Boulder, CO, USA. [2]Department of Biological Sciences and Institute of Molecular Biophysics, Florida State University, Tallahassee, FL, USA. [3]Department of Medicine, University of Colorado Anschutz Medical Campus, Aurora, CO, USA. [4]Present address: Department of Biological Sciences, University of Southern California, Los Angeles, CA, USA. ✉e-mail: yin@bio.fsu.edu; jingshi.shen@colorado.edu

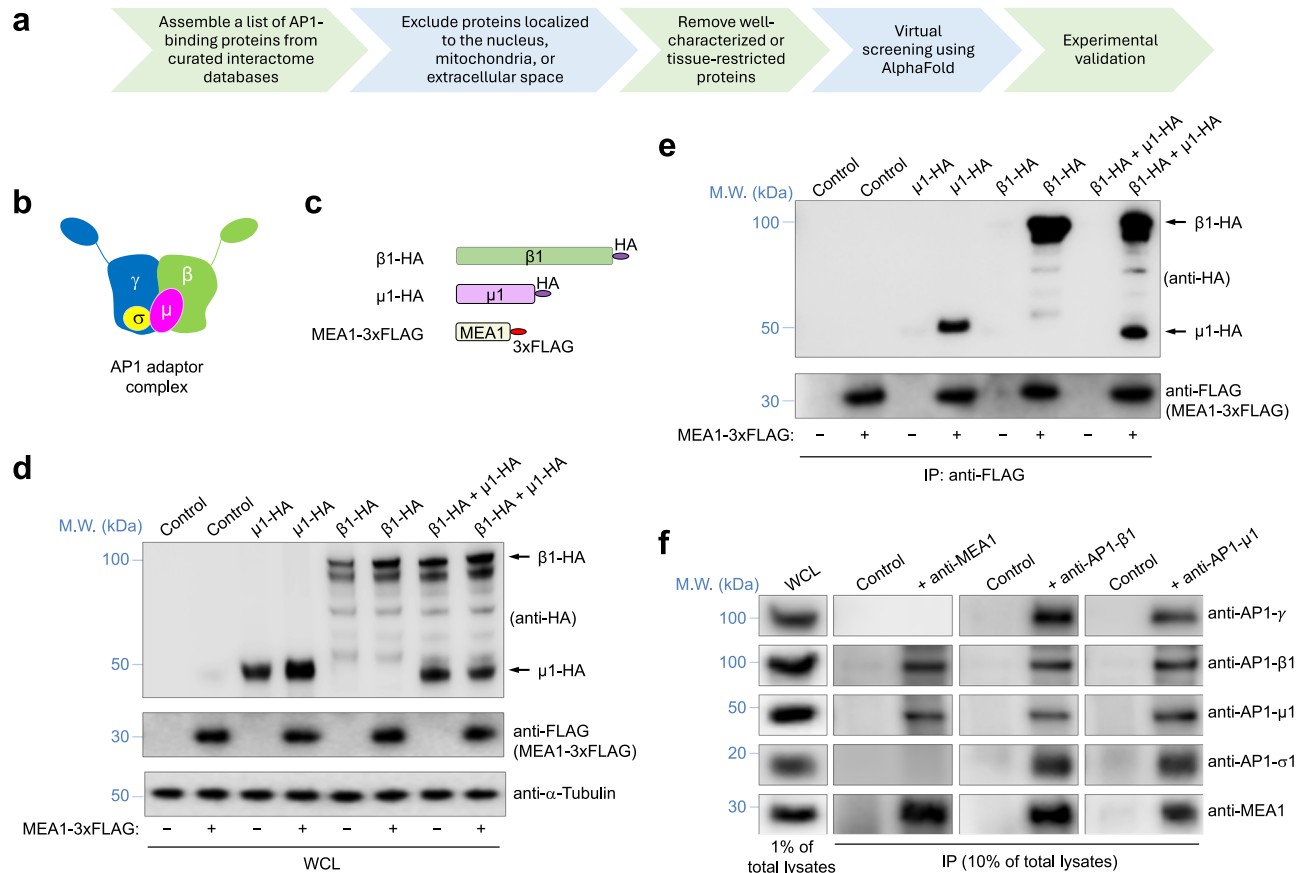

**Fig. 1 | Identification of MEA1 as an AP1-interacting protein. a** Procedure for identifying AP1-interacting proteins through proteomic and structure-based screening. In the AlphaFold-based virtual screen, an ipTM score of 0.5 was used as a preliminary cutoff for experimental validation. Candidate proteins interacting with human AP1 subunits are listed in Supplementary Data 1. **b** Illustration of the heterotetrameric AP1 adaptor complex. **c** Diagrams of HA-tagged β1 and μ1, and 3 × FLAG-tagged MEA1. **d** Representative immunoblots from three independent experiments showing the indicated proteins in whole-cell lysates (WCL). 3 × FLAG-MEA1 was transiently expressed in HEK 293 T cells with either an empty vector (control) or plasmids encoding HA-tagged AP1 subunits. M.W., molecular weight.

**e** Representative immunoblots from three independent experiments showing the interaction of 3 × FLAG-tagged MEA1 with HA-tagged AP1 subunits corresponding to WCL samples in (**d**). MEA1 was immunoprecipitated using anti-FLAG antibodies, and proteins in the immunoprecipitates were detected by immunoblotting. **f** Representative immunoblots from three independent experiments showing interactions between endogenous MEA1 and AP1 subunits. Endogenous MEA1, μ1, and β1 were individually immunoprecipitated from HEK 293 T cells using specific antibodies and protein A/G beads. Proteins in the immunoprecipitates were detected by immunoblotting.

aggregation-prone subunits or assembly intermediates, shield exposed surfaces, and prevent their degradation[30–35]. For AP1, Alpha- and Gamma-Adaptin Binding Protein (AAGAB) plays a central role[25,26,28–30,36]. AAGAB functions as a bi-handed assembly chaperone, using its GTPase-like domain and C-terminal domain to bind and stabilize the σ1 and γ subunits, respectively, and promote their pairing into a γ:σ1 hemicomplex, a key intermediate en route to the mature adaptor[25–30]. Upon formation of the full AP1 tetramer, AAGAB dissociates, distinguishing it from downstream effectors such as clathrin and cargo adaptors, which associate only with the fully assembled complex[27–30]. Beyond AP1, AAGAB also contributes to the assembly of AP2 and AP4, which participate in cargo transport from the plasma membrane and the TGN, respectively[24–30].

In this work, we sought to identify additional regulators of AP1 using integrated interactome screens. These analyses revealed Male-Enhanced Antigen 1 (MEA1), a previously uncharacterized protein, as a second assembly chaperone for AP1. MEA1 loss resulted in depletion of AP1 subunits, impaired trafficking of AP1-dependent cargoes, and defective termination of STING signaling. Mechanistically, MEA1 is a bi-handed chaperone that binds and stabilizes μ1 and β1, forming a complementary hemicomplex to the AAGAB-stabilized γ:σ1 pair. Collision of these MEA1- and AAGAB-bound intermediates produces the

full AP1 tetramer and triggers release of both chaperones. These findings establish MEA1 as a key regulator of AP1 biogenesis and reveal a dual-chaperone collision mechanism that may represent a general principle in multisubunit complex assembly.

## Results

### Proteomic and virtual interactome screens identify MEA1 as an AP1-binding protein

To identify previously unrecognized regulators of AP1-dependent trafficking, we implemented a two-stage discovery strategy combining proteomic analysis with AlphaFold-based virtual interaction screening (Fig. 1a). We first assembled a list of candidate AP1-interacting proteins from the Biological General Repository for Interaction Datasets (Bio-GRID), which compiles protein–protein interactions from both high-throughput studies and focused experiments[37]. This initial list included 489 proteins reported to interact with at least one human AP1 subunit (Supplementary Data 1). To enrich for biologically relevant candidates, we excluded proteins annotated as localized to compartments not associated with AP1 function, including the nucleus, extracellular space, and mitochondria. We further refined the list by removing proteins with restricted tissue distributions (based on the Human Protein Atlas) and those with well-established functions. The resulting

candidate set was analyzed using AlphaFold-based virtual screening to identify proteins predicted to form interfaces with AP1 subunits, which are indicative of potential direct AP1 regulators (Fig. 1a).

A strong candidate identified by this integrated screening approach is MEA1, an uncharacterized soluble protein of ~20 kDa. MEA1 passed all initial filtering criteria, appeared in the interactomes of both μ1 and β1 (Supplementary Data 1), and AlphaFold predicted protein complexes between MEA1 and these subunits with high ipTM scores (Supplementary Data 1). The convergence of these independent criteria distinguished MEA1 from other candidates and prompted us to investigate it further. Although initially cloned from a testis cDNA library (hence its name)[38], MEA1 is ubiquitously expressed across tissue types and is detected in all examined human cell lines[39,40], indicating a general cellular role. MEA1 is conserved across all vertebrates and is present in some invertebrates such as *Drosophila melanogaster* (Supplementary Fig. 1). MEA1 shares no sequence similarity with known proteins and is predicted to be an intrinsically disordered protein containing short α-helical segments but no stable tertiary structure (Supplementary Fig. 1b and AlphaFold Database, #AF-Q16626-F1).

To validate the interaction between MEA1 and AP1 subunits, we performed co-immunoprecipitation (co-IP) assays using human cell extracts. We observed that 3 × FLAG-tagged MEA1 co-immunoprecipitated with HA-tagged μ1 and β1 subunits, both individually and in combination (Fig. 1b–e). Binding between endogenous MEA1 and AP1 subunits was also confirmed by co-IP using antibodies against MEA1, μ1, or β1 (Fig. 1f). Corroborating the results of the interactome screens, these data demonstrate that MEA1 associates with AP1 subunits in the cell.

## MEA1 is required for AP1-dependent cargo trafficking and termination of STING signaling

To determine whether MEA1 is required for AP1 function, we first examined the trafficking of folate receptor alpha (FOLR1), a cargo protein whose surface levels increase when AP1-dependent endolysosomal sorting is disrupted (Fig. 2a)[29]. We observed that *MEA1* KO cells showed a markedly increased surface FOLR1 level, as measured by flow cytometry (Fig. 2b, c). As a control, we included *AP1G1* KO cells, in which the γ subunit of AP1 was deleted. As previously reported[29], *AP1G1* KO cells exhibited elevated surface FOLR1 expression (Fig. 2c). These results suggest that MEA1 is required for AP1-mediated trafficking of constitutive cargoes.

We next examined how MEA1 regulates signal-stimulated cargoes. Termination of STING signaling depends on AP1-mediated packaging of phosphorylated STING into CCVs for lysosomal degradation (Fig. 2a)[23]. Cells were treated with diABZI, a potent STING agonist[41], and pathway activation was assessed by immunoblotting (Fig. 2d). In both *MEA1* and *AP1G1* KO cells, levels of phosphorylated STING and phosphorylated TBK1 were markedly elevated compared to WT cells (Fig. 2d, e), consistent with defective STING clearance and prolonged signaling. To determine whether this defect reflected impaired sorting into vesicles, we isolated CCVs and analyzed their contents. In WT cells, phosphorylated STING and phosphorylated TBK1 were readily detected in CCVs (Fig. 2f). In contrast, both were strongly reduced in CCVs from *MEA1* or *AP1G1* KO cells (Fig. 2f, g), indicating a defect in AP1-mediated cargo incorporation. Thus, MEA1 is required for AP1-dependent termination of STING signaling.

Together, these results demonstrate that MEA1 is critical for the trafficking of both constitutive and signal-activated AP1 cargoes. The similarity between *MEA1* and *AP1G1* KO phenotypes supports a model in which MEA1 is required for AP1 function.

## Loss of AP1 adaptor in MEA1-deficient cells

To investigate the molecular mechanism by which MEA1 regulates AP1-dependent trafficking, we examined the expression and localization of AP1 in *MEA1* KO cells. Interestingly, immunoblotting revealed that

all four AP1 subunits—γ, β1, μ1, and σ1—were strongly reduced in *MEA1* KO cells compared to WT controls (Fig. 3a, b). Re-expression of MEA1 fully restored AP1 subunit levels (Supplementary Fig. 2a, b), confirming that the AP1 reduction was specifically due to MEA1 loss. Similarly, all four AP1 subunits were depleted in *AP1G1* KO cells (Fig. 3a, b), consistent with the interdependent nature of adaptor complex subunits, where loss of one subunit leads to degradation of the others[27–30]. The comparable phenotypes in *MEA1* and *AP1G1* KO cells suggest that MEA1 is required for maintaining AP1 complex integrity.

Next, we visualized AP1 in WT and *MEA1* KO cells using confocal imaging. Because subunits of adaptor protein complexes exist primarily as part of full tetrameric complexes in the cell[1,29,30,42], staining of individual AP1 subunits, such as γ reflects the abundance and localization of full AP1 adaptor. In WT cells, staining for γ revealed prominent perinuclear puncta (Fig. 3c, d), consistent with membrane-associated AP1 on the TGN and endosomal compartments. These AP1 puncta were markedly reduced in *MEA1* KO cells (Fig. 3c, d), confirming the loss of AP1 subunits in MEA1-deficient cells.

To evaluate the specificity of MEA1, we examined its relationship with other adaptor protein complexes. Mammalian cells express five heterotetrameric adaptor protein complexes (AP1–AP5), which share a conserved architecture[1,24,29,30]. These adaptor complexes, however, function in distinct trafficking pathways and their subunits are generally not interchangeable[1,29]. Besides μ1 and β1, the interactome of MEA1 also included the β2 and μ2 subunits of AP2 (BioGRID)[37], a central regulator of clathrin-mediated endocytosis[18,43–47]. The interactions of MEA1 with these AP2 subunits were confirmed using co-IP (Supplementary Fig. 3a, b). However, AP2 subunit levels were only moderately reduced in *MEA1* KO cells (Supplementary Fig. 3c, d), in contrast to the strong depletion of AP1 (Fig. 3), suggesting that MEA1 plays a limited role in AP2 regulation. Expression levels of AP3, AP4, and AP5 remained unchanged (Supplementary Fig. 3c, d), consistent with the absence of MEA1 in their interactomes (BioGRID)[37].

These results demonstrate that MEA1 is critical for maintaining the integrity of the AP1 adaptor complex, and suggest that its function is primarily dedicated to AP1, without broadly regulating adaptor protein complexes.

## MEA1 selectively binds and stabilizes the μ1 and β1 subunits of AP1

The coordinated loss of all AP1 subunits in *MEA1* KO cells is reminiscent of the phenotype observed in AAGAB-deficient cells, in which AP1 subunits fail to assemble and are consequently degraded[24,27,28,30]. This similarity led us to hypothesize that MEA1 functions as another assembly chaperone for AP1. Proteomic interactome data revealed that MEA1 associates with μ1 and β1, but not with γ or σ1 (Supplementary Data 1). To validate this binding pattern, we performed co-IP experiments in human cells. MEA1 co-immunoprecipitated with μ1 and β1, but not with γ or σ1 (Fig. 4a–c). By contrast, AAGAB bound to γ and σ1, but not μ1 or β1 (Fig. 4a–c). These data suggest that MEA1 selectively binds μ1 and β1 prior to full tetramer formation, consistent with the behavior of assembly chaperones.

To determine whether MEA1 directly binds to μ1 and β1 and stabilizes them, we co-expressed these proteins in *E. coli*, which lack endogenous AP1 and MEA1. When expressed alone, little soluble μ1 and β1 were obtained (Fig. 4d, e), consistent with the inherent instability of unassembled subunits of adaptor protein complexes[27–30]. However, co-expression with MEA1 markedly increased soluble μ1 and β1 proteins isolated from *E. coli*, and MEA1 was found in complexes with these AP1 subunits (Fig. 4d, e). These results demonstrate that MEA1 directly binds to μ1 and β1, and stabilizes them in a soluble state. This behavior is a hallmark of assembly chaperones, which prevent misfolding and aggregation by shielding exposed hydrophobic surfaces that are normally buried within the assembled complex[34,48,49]. Together, these

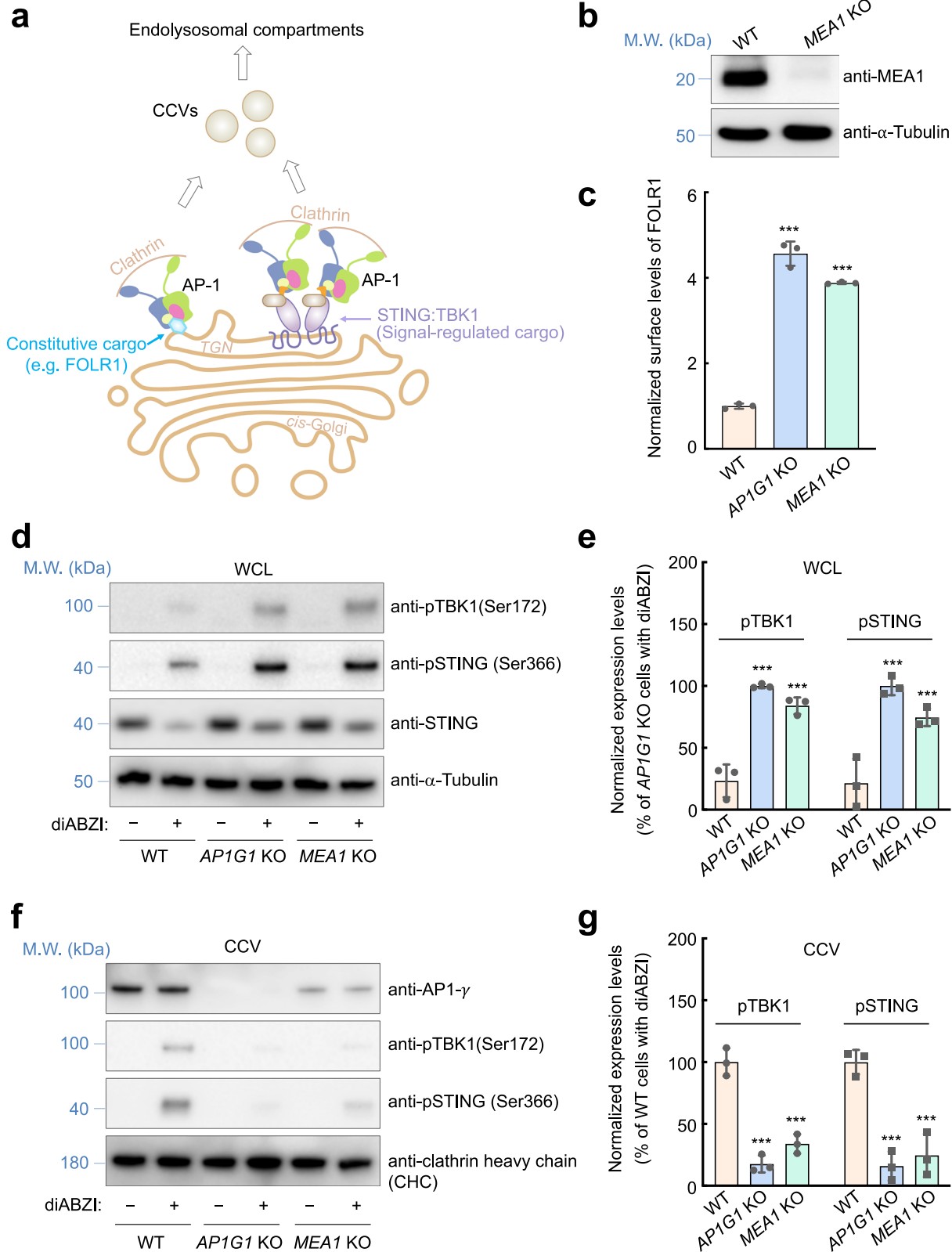

findings support a model in which MEA1 functions as a μ1- and β1-specific assembly chaperone for the AP1 adaptor complex.

## MEA1 is a bi-handed assembly chaperone possessing two AP1-recognizing domains

To understand how MEA1 recognizes μ1 and β1, we used AlphaFold to predict the configuration of MEA1:AP1 complexes. Using μ1, β1, and full-length (FL) MEA1 as input, AlphaFold predicted a high-confidence trimeric structure in which the N-terminal domain (NTD) of MEA1 interacts with μ1, while the CTD binds β1 (Supplementary Fig. 4 and Supplementary Data 2). To test this architecture, we performed structural predictions using the NTD and CTD of MEA1 individually with μ1 and β1. AlphaFold predicted high-confidence structures for MEA1-NTD:μ1 and MEA1-CTD:β1 dimers (Fig. 5a–d and Supplementary

**Fig. 2 | MEA1 is required for AP1-dependent cargo trafficking and termination of STING signaling. a** Diagram illustrating AP1-dependent transport of constitutive and signal-induced cargoes such as FOLR1 and STING from the TGN to endolysosomal compartments. Impairment of AP1 function results in aberrant accumulation of FOLR1 on the cell surface and reduced termination of STING signaling. **b** Representative immunoblots from three independent experiments showing the expression of the indicated proteins in WT and *MEA1* KO HeLa cells. **c** Normalized surface levels of FOLR1 measured using flow cytometry. Data normalization was performed by setting the mean value of WT data points as 1 and all data points were normalized to that mean value. Data are presented as mean ± SD of three biological replicates. ***$P < 0.001$ (one-way ANOVA). **d** Representative immunoblots showing the total levels of the indicated proteins in WT or KO RPE1 cells, a non-transformed

human cell line with a near-diploid genome[65]. Cells were treated with diABZI for two hours before harvest. **e** Quantification of immunoblot data, including those shown in (**d**). Data normalization was performed by setting the mean value of *AP1G1* KO data points as 100% and all data points, including *AP1G1* KO ones were normalized to that mean value. Data are presented as mean ± SD of three biological replicates. ***$P < 0.001$ (one-way ANOVA). **f** Representative immunoblots showing the levels of the indicated proteins in CCVs isolated from WT or KO RPE1 cells. Cells were cultured and stimulated as in (**d**). **g** Quantification of immunoblot data, including those shown in (**f**). Data normalization was performed by setting the mean value of WT data points as 100% and all data points including WT ones were normalized to that mean value. Data are presented as mean ± SD of three biological replicates. ***$P < 0.001$ (one-way ANOVA).

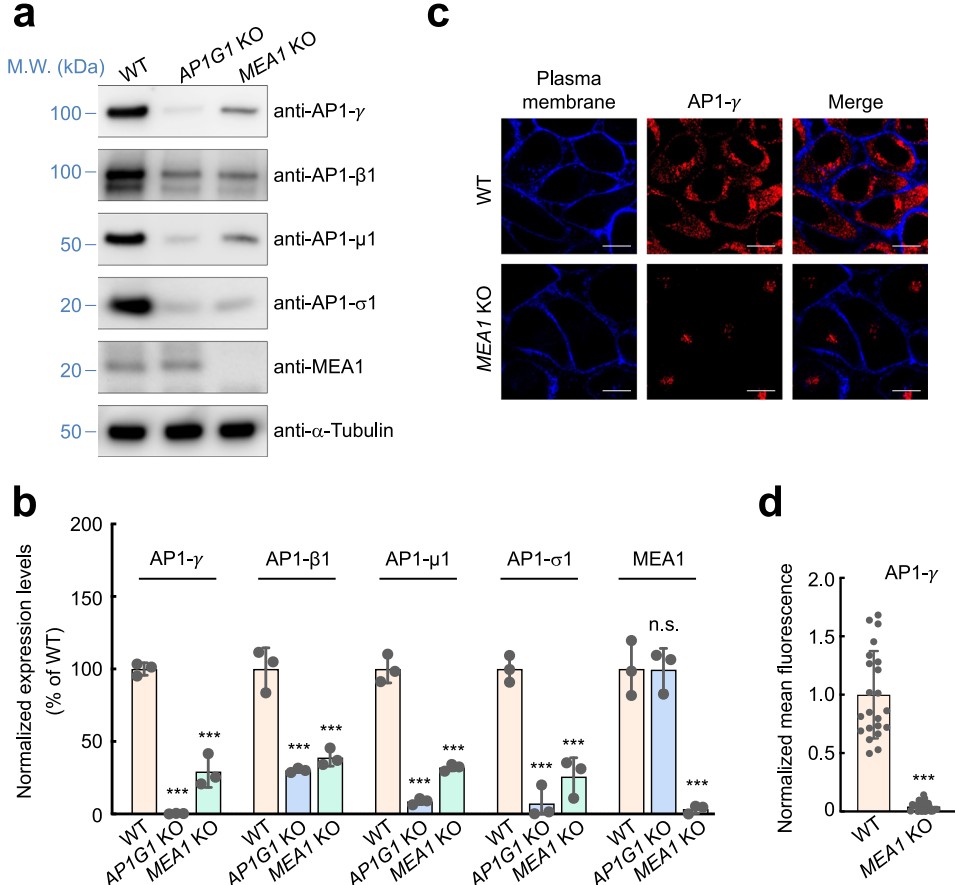

**Fig. 3 | Loss of AP1 in *MEA1* KO cells. a** Representative immunoblots showing the expression of the indicated proteins in WT and KO RPE1 cells. **b** Quantification of protein expression in WT and KO RPE1 cells based on immunoblots. In this figure, data normalization was performed by setting the mean value of WT data points as 100% or 1, and all data points were normalized to that mean value. Data are presented as mean ± SD of three biological replicates. ***$P < 0.001$; n.s., $P > 0.05$ (two-way ANOVA). **c** Representative confocal images of AP1 stained with antibodies recognizing γ in WT and *MEA1* KO HeLa cells. The plasma membrane was stained with CF405-conjugated Concanavalin A. Scale bars, 10 μm. **d** Quantification of γ staining based on confocal images. Images were captured as in (**c**), and analyzed using ImageJ. Each dot represents imaging data of an individual cell. Error bars indicate SD ($n = 22$). ***$P < 0.001$ (two-sided Student's *t*-test).

Fig. 5, and Supplementary Data 3 and 4), but not for the mismatched pairs (MEA1-NTD:β1 or MEA1-CTD:μ1). These observations suggest that MEA1 uses its NTD and CTD to recognize μ1 and β1, respectively (Fig. 5e). AlphaFold also predicted similar binding patterns of MEA1 to the μ2 and β2 subunits of the AP2 adaptor (Supplementary Fig. 6 and Supplementary Data 5), consistent with a conserved mode of substrate recognition.

To test these predicted binding modes, we performed binding assays with recombinant proteins expressed in *E. coli.* Pull-down assays showed that MEA1-NTD bound and stabilized μ1 to a similar extent as FL MEA1, whereas MEA1-CTD neither bound nor stabilized μ1 (Fig. 5f and Supplementary Fig. 7a). Conversely, MEA1-CTD, but not NTD,

bound and stabilized β1 (Fig. 5g and Supplementary Fig. 7b). These results were recapitulated in human cells: co-IP experiments revealed that MEA1-NTD associated with μ1 but not β1, while MEA1-CTD bound to β1 but not μ1 (Supplementary Fig. 8). Together, these assays validate the interaction modes of the MEA1-NTD:μ1 and MEA1-CTD:β1 complexes predicted by AlphaFold.

According to the AlphaFold-predicted structure of the MEA1-NTD:μ1 complex, Y70 of MEA1 forms hydrogen bonds with D174 and R201 of μ1, while Q71 and L73 of MEA1 form hydrogen bonds with V409 and P407 of μ1, respectively (Fig. 5a, b and Supplementary Data 3). At the binding interface of the MEA1-CTD:β1 complex, K142 of MEA1 forms hydrogen bonds with T532 of β1 (Fig. 5c, d and Supplementary

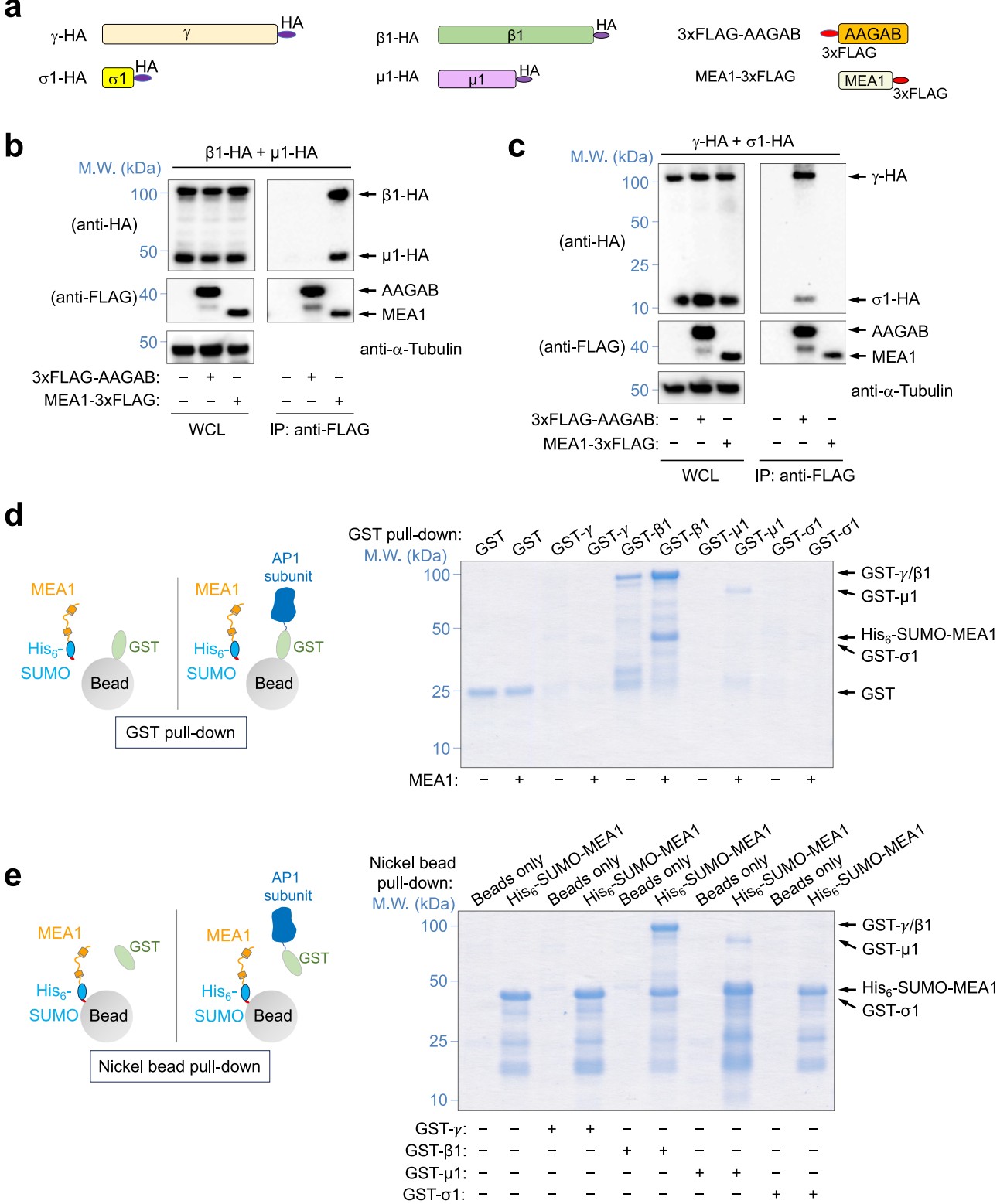

**Fig. 4 | MEA1 binds and stabilizes the μ1 and β1 subunits of AP1. a** Diagrams of HA-tagged AP1 subunits and 3 × FLAG-tagged MEA1 and AAGAB used in co-IP experiments. **b, c** Representative immunoblots from three independent experiments showing the interaction of 3 × FLAG-tagged MEA1 and AAGAB with HA-tagged AP1 subunits. The 3 × FLAG-MEA1 and 3 × FLAG-AAGAB proteins were transiently expressed in HEK 293 T cells along with HA-tagged AP1 subunits. MEA1 and AAGAB were immunoprecipitated from cell lysates using anti-FLAG antibodies, and proteins in the immunoprecipitates were detected by immunoblotting. **d, e** Left: diagrams of GST and nickel bead pull-down assays. Right: representative Coomassie blue-stained gels from three independent experiments showing MEA1-mediated binding and stabilization of AP1 subunits in these assays. GST and GST-tagged AP1 subunits (γ [core domain, a.a. 1–595], β1 [core domain, a.a. 1–584], FL μ1, and FL σ1) were individually co-expressed with an empty vector or a plasmid encoding His$_6$-SUMO-tagged MEA1 in *E. coli*. Proteins were isolated using glutathione beads (**d**) or nickel beads (**e**).

Data 4). V141 and M145 of MEA1 insert into the continuous hydrophobic groove mainly formed by F492, L493, V503, L507, W527, and L530 of β1, and T144 of MEA1 engages in a polar interaction with Q500 of β1 (Fig. 5c, d and Supplementary Data 4). These MEA1:AP1 binding interfaces are consistent with those predicted in the structural model using FL MEA1 (Supplementary Fig. 4 and Supplementary Data 2). Together, these structural models suggest that MEA1 binding may help shield exposed hydrophobic residues on μ1 and β1 (Supplementary Fig. 5c, f).

The structural models of the MEA1:AP1 complexes showed strong interface predicted Template Modeling (ipTM) scores (≥ 0.75), indicative of accurate structural predictions[50–54], and their predicted domain-specific binding modes were experimentally validated (Fig. 5f, g and Supplementary Figs. 7 and 8). Moreover, the molecular interactions in these structural models are mediated by evolutionarily conserved residues of MEA1 (Fig. 5a–d and Supplementary Figs. 1 and 4a, b), supporting their functional relevance. To further probe the biological relevance of the structural models, we introduced point mutations into conserved MEA1 residues predicted to interact with μ1 or β1. Mutation of Leu73, a residue within the μ1-binding interface of MEA1-NTD (Fig. 5b and Supplementary Fig. 9a), significantly reduced MEA1:μ1 binding (Supplementary Fig. 9b). Next, we mutated the VKTM motif (V141/K142/T144/M145) in MEA1, which contacts β1 (Fig. 5d and Supplementary Fig. 10a). We observed that mutations of this VKTM motif diminished the binding of MEA1 to β1 (Supplementary Fig. 10b). A single V141A substitution also markedly reduced the MEA1:β1 interaction (Supplementary Fig. 10). Next, we expressed these MEA1 mutants in *MEA1* KO cells. We observed that these mutants failed to restore AP1 expression or proper STING signaling when expressed at levels comparable to WT MEA1 (Supplementary Fig. 11). These results validate the structural models and pinpoint key residues required for MEA1:AP1 interactions.

Together, these findings demonstrate that MEA1 is a bi-handed assembly chaperone, with two functionally distinct domains that independently recognize μ1 and β1. They also illustrate the accuracy of AlphaFold in predicting biologically meaningful structures of MEA1:AP1 complexes.

### MEA1 and AAGAB cooperatively control AP1 assembly and dissociate upon formation of the full AP1 adaptor

Because MEA1 and AAGAB recognize distinct halves of the AP1 adaptor (Fig. 4b, c)[26–30], we posit that they act cooperatively to promote AP1 assembly. To test this possibility, we reconstituted the chaperone-assisted AP1 assembly process in vitro using recombinant proteins. The GST-γ:σ1:AAGAB trimer was expressed and purified from *E. coli* and incubated with *E. coli* lysates expressing MEA1, μ1, and β1 (Fig. 6a). Using a GST pull-down assay, we observed that both μ1 and β1 were efficiently recruited to GST-γ:σ1 immobilized on glutathione beads (Fig. 6b, c), consistent with formation of the full AP1 complex. Notably, this recruitment was accompanied by the release of both MEA1 and AAGAB (Fig. 6b, c), indicating that AP1 tetramer formation is coupled to chaperone release. MEA1 and AAGAB did not interact directly when tested using recombinant proteins (Supplementary Fig. 12), and they are absent from each other's interactomes (BioGRID)[37], suggesting that these chaperones do not stably exist in the same protein complex. Moreover, AAGAB levels were unchanged in *MEA1* KO cells, and MEA1 levels were unchanged in *AAGAB* KO cells (Supplementary Fig. 2c, d), indicating that neither chaperone is required for the stability of the other. These findings support a dual-chaperone collision model, in which MEA1:μ1:β1 and AAGAB:γ:σ1 collide without stably associating with each other, and both MEA1 and AAGAB are released upon formation of the full AP1 tetramer (Fig. 6d).

A prediction of this dual-chaperone collision model is that MEA1 and AAGAB function in the cytosol, where AP1 assembles, and do not associate with membrane-bound AP1 in cells. At steady state, AP1 subunits are typically incorporated into the full tetrameric complex and localize to membrane compartments[1,12–14], whereas assembly chaperones are expected to dissociate upon AP1 assembly and remain cytosolic. Confocal imaging revealed that both MEA1 and AAGAB displayed diffuse cytosolic localization (Fig. 6e), distinct from the membrane-associated pattern of AP1 (Fig. 3c). Super-resolution imaging further showed no significant co-localization between MEA1 and AP1 puncta (Fig. 6f–h). Consistent with these imaging results, cell fractionation showed that the transferrin receptor (TfR) and AP1 γ subunit were exclusively in the membrane fraction, whereas MEA1 and AAGAB were confined to the soluble cytosolic fraction along with glyceraldehyde 3-phosphate dehydrogenase (GAPDH), a cytosolic marker (Fig. 6i). While these findings cannot fully exclude the presence of membrane-associated assembly intermediates, they support the conclusion that MEA1 and AAGAB dissociate after AP1 assembly and do not accompany the mature adaptor to membrane compartments. In agreement with this, structural modeling showed that MEA1 occupies the γ-binding site on β1 (Supplementary Fig. 13), indicating that MEA1 must be released before γ can associate with β1 to form the full AP1 adaptor.

Together, these findings demonstrate that the bi-handed chaperones MEA1 and AAGAB act in concert to promote AP1 assembly by stabilizing separate subunit pairs, colliding to form the AP1 tetramer, and dissociating upon full AP1 formation (Fig. 7).

## Discussion

In this study, we identified MEA1 as a critical regulator of the AP1 adaptor complex. MEA1 is a bi-handed assembly chaperone that acts in concert with another bi-handed chaperone, AAGAB, to drive AP1 assembly. While adaptor protein complexes can assemble spontaneously in heterologous systems[6,55], their formation in native cellular environments requires dedicated assembly chaperones[24,28–30]. MEA1 contains two distinct substrate-binding domains that engage and stabilize the μ1 and β1 subunits, whereas AAGAB binds and stabilizes γ and σ1. Together, these chaperones promote AP1 assembly through a dual-chaperone collision mechanism: when the MEA1:μ1:β1 complex encounters the AAGAB:γ:σ1 complex, the full AP1 tetramer forms, triggering the release of both chaperones (Fig. 7). In the absence of either MEA1 or AAGAB, AP1 fails to assemble, leading to degradation of all AP1 subunits. Chaperone-assisted AP1 assembly is ATP-independent and driven solely by protein-protein interactions. Both MEA1 and AAGAB are cytosolic and excluded from membrane-bound AP1, consistent with cytosolic assembly of adaptor complexes and membrane recruitment only after tetramer formation. Nevertheless, we cannot exclude the possibility that chaperone-bound AP1 intermediates may transiently associate with membranes during assembly.

As bi-handed assembly chaperones, MEA1 and AAGAB play two key roles in AP1 assembly: (1) they bind and stabilize two subunits in a soluble, assembly-competent state while shielding them from degradation by quality control pathways targeting exposed hydrophobic interfaces, and (2) they promote subunit pairing into a hemicomplex via a spatial proximity effect, a critical step toward formation of the full AP1 complex. Our data support the existence of transient μ1:β1 and γ:σ1 hemicomplexes in vivo, whose stability depends on MEA1 and AAGAB, respectively. A key future direction is to determine how these chaperones promote hemicomplex formation and pairing to generate full AP1. It will also be important to understand how AP1 assembles in single-celled eukaryotes such as yeast, which express AP1 but lack MEA1 or AAGAB (Supplementary Fig. 1)[1,2]. Although the yeast protein Irc6 resembles the GTPase-like domain of AAGAB[56], it lacks the γ-binding CTD and is therefore unlikely to act as an AP1 assembly chaperone[26,28]. Such organisms, with simpler proteomes, may assemble AP1 using only folding chaperones, or they may rely on functionally equivalent assembly chaperones that lack detectable sequence similarity to MEA1 or AAGAB.

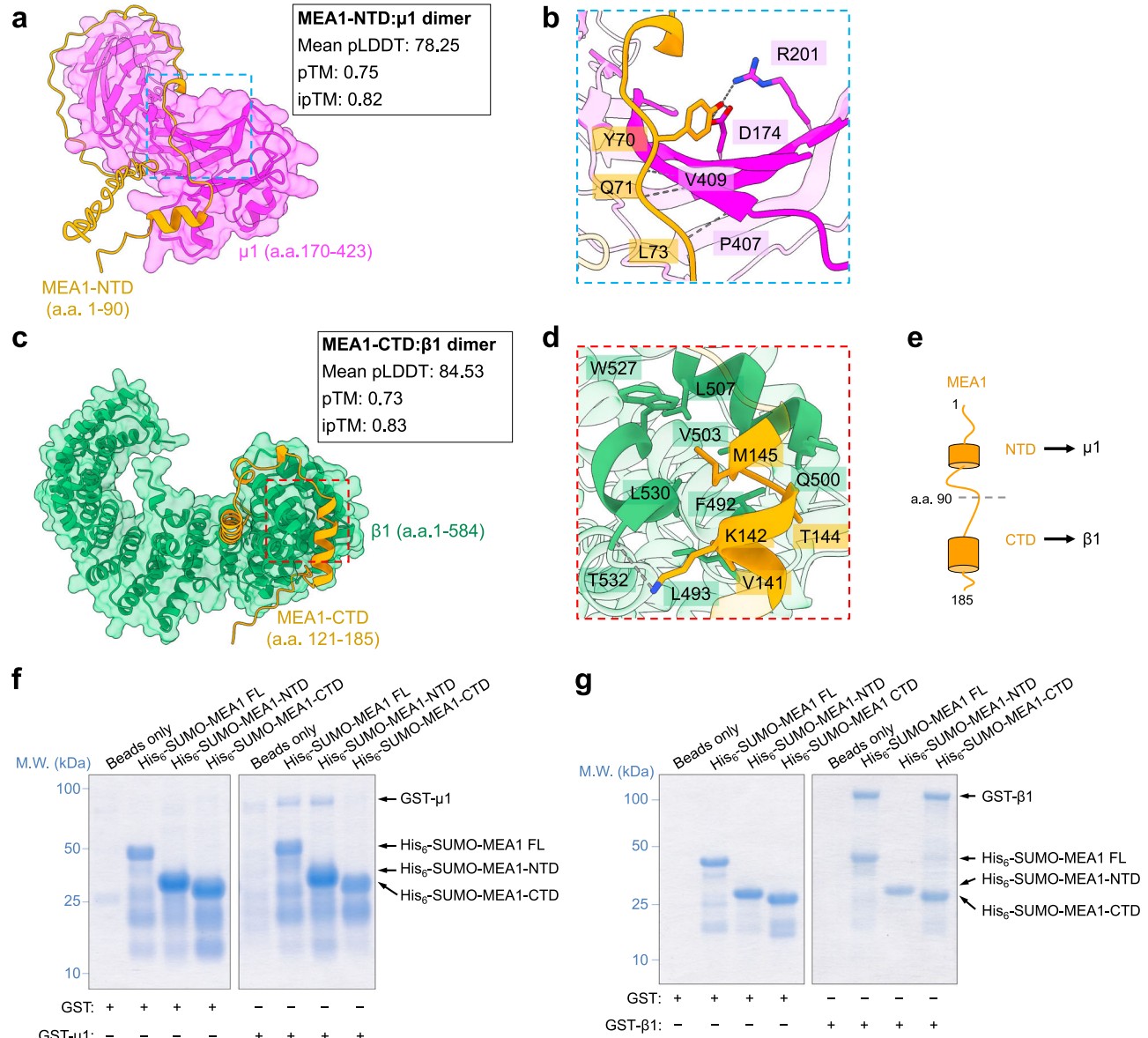

**Fig. 5 | MEA1 is a bi-handed chaperone with two AP1-recognizing domains.**
**a** AlphaFold-predicted structure of the MEA1-NTD:μ1 dimer using MEA1-NTD (a.a. 1–90) and the a.a. 170–423 of μ1 as input. The CIF file of the predicted structure is included in Supplementary Data 3. **b** Binding interface between μ1 and MEA1-NTD, corresponding to the dashed box shown in (**a**). **c** AlphaFold-predicted structure of the MEA1-CTD:β1 dimer using MEA1-CTD (a.a. 121–185) and the core domain of β1 (a.a. 1–584) as input. The CIF file of the predicted structure is included in Supplementary Data 4. **d** Binding interface between β1 and MEA1-CTD (a.a. 121–185),

corresponding to the dashed box shown in (**c**). **e** Diagram illustrating the predicted binding model of MEA1 NTD and CTD to μ1 and β1 subunits. **f, g** Representative Coomassie blue-stained gels from three independent experiments showing MEA1:AP1 subunit interactions in pull-down assays performed using a setup similar to that in this figure (**e**). GST and GST-tagged FL μ1 (**f**) or the core domain of β1 (**g**) were individually co-expressed with His₆-SUMO-tagged MEA1 (FL, NTD, or CTD) in *E. coli*. Proteins were isolated using nickel beads.

Besides AP1, MEA1 also binds the β2 and μ2 subunits of AP2, although its effect on AP2 formation is relatively modest (Supplementary Fig. 3), indicating a limited, modulatory role in AP2 assembly. By contrast, AP2 assembly depends strongly on AAGAB and CCDC32, the latter being specific to AP2 and not involved in AP1 formation[27]. CCDC32 forms a trimer with α and σ2 that can recruit μ2, generating an assembly intermediate capable of nucleating the full AP2 complex, including β2 incorporation[27]. In this context, MEA1 may facilitate AP2 assembly by stabilizing μ2 and β2, but it is not essential. This contrasts with AP1 assembly, in which MEA1 is indispensable for stabilizing μ1 and β1, which are not directly recruited by the AAGAB:γ:σ1 complex. These results suggest that AP1 lacks an equivalent of CCDC32, placing greater reliance on MEA1

as a bi-handed assembly chaperone. They also indicate that the AAGAB:α:σ2 trimer cannot effectively collide with a MEA1:μ2:β2 trimer to form the AP2 adaptor complex, necessitating an AAGAB-to-CCDC32 handover[27]. Thus, despite their shared architecture, AP1 and AP2 employ distinct chaperone networks and assembly mechanisms, likely reflecting differences in subunit interfaces and recruitment dynamics.

To date, three assembly chaperones have been identified for adaptor protein complexes: AAGAB, MEA1, and CCDC32. AAGAB promotes the assembly of AP1, AP2, and AP4; MEA1 is critical for AP1 and plays a limited role in AP2 assembly; and CCDC32 functions specifically in AP2 assembly[24,27–30]. Together, these factors define a mechanistic pathway we term Chaperone-assisted Adaptor Protein Assembly

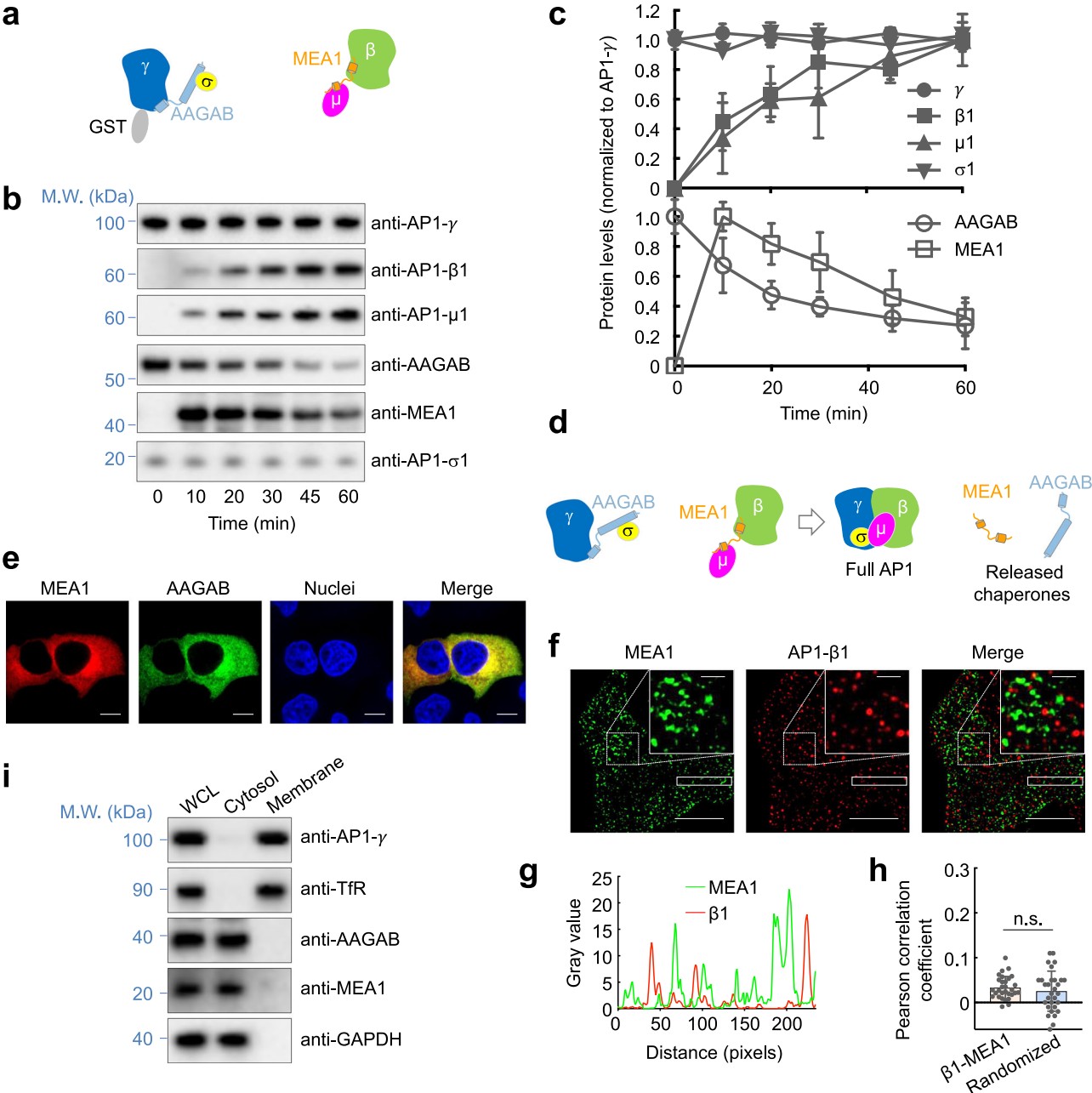

**Fig. 6 | MEA1 acts in concert with AAGAB to regulate AP1 assembly. a** Diagram of the GST pull-down assay. The GST-γ:σ1:AAGAB ternary complex (core domain of γ, FL σ1, and His₆-SUMO-tagged FL AAGAB) was expressed and isolated from *E. coli* using glutathione beads. The GST-γ:σ1:AAGAB complex bound to glutathione beads was added to *E. coli* lysates expressing β1 (core domain), FL μ1, and His₆-SUMO-tagged FL MEA1. After incubation for the indicated periods at 37 °C, the glutathione beads were washed and proteins bound to the beads were detected by immunoblotting. **b** Representative immunoblots showing the amounts of proteins bound to the glutathione beads. **c** Quantification of proteins based on immunoblots from three independent experiments. Data normalization was performed by setting the mean value of GST-γ at 0 min as 1 and all data points were normalized to that mean value. Error bars indicate SD ($n = 3$ biological replicates). **d** Diagram depicting the dissociation of MEA1 and AAGAB from AP1 upon formation of full AP1. For clarity, only the core domains of γ and β1 are shown. **e** Representative confocal images ($n = 25$) showing the cytosolic localization of MEA1 and AAGAB. ALFA-tagged MEA1 was labeled using anti-ALFA nanobodies fused to mScarlet. The

3 × FLAG-tagged AAGAB was labeled using anti-FLAG antibodies and Alexa Fluor 488-conjugated secondary antibodies. Nuclei were stained using Hoechst 33342. Scale bars, 10 μm. **f** Representative SIM images ($n = 30$) showing the subcellular localization of MEA1 and β1. The 3 × FLAG-tagged MEA1 was labeled using Alexa Fluor 488-conjugated anti-FLAG antibodies, whereas endogenous β1 was stained using anti-β1 antibodies and Alexa Fluor 568-conjugated secondary antibodies. Scale bars, 10 μm for main images and 1 μm for enlarged views. **g** Profile analysis plot comparing the distributions of MEA1 and β1 within the rectangular areas shown in (**f**). **h** Quantification of MEA1 co-localization with AP1 adaptor (β1 staining) using the Pearson correlation coefficient. Images were acquired as in (**f**) and analyzed using ImageJ. Each dot represents an individual cell. For randomized control data, β1 images were rotated 90° while MEA1 images remained unrotated. Error bars indicate SD ($n = 30$). n.s., $P > 0.05$ (two-sided Student's *t*-test). **i** Representative immunoblots from three independent experiments showing the levels of the indicated proteins in WCL, cytosolic, and membrane fractions.

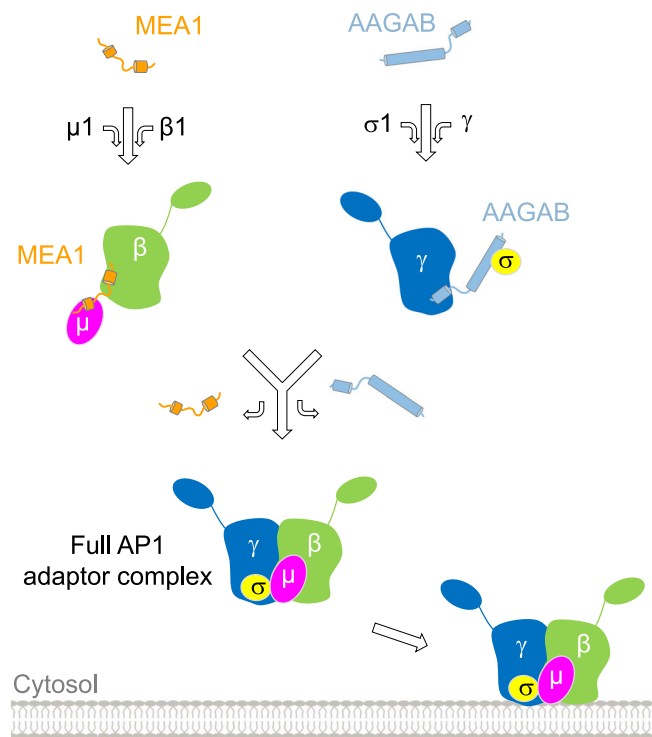

**Fig. 7** | A dual-chaperone collision model of AP1 assembly.

(CAPA), which ensures the efficient and accurate formation of adaptor protein complexes.

Comparative analysis of CAPA chaperones reveals several emerging principles. First, bi-handed chaperones are central to adaptor complex assembly, simultaneously engaging and stabilizing two distinct subunits. Because free subunits are often unstable and short-lived, random encounters are inefficient; bi-handed chaperones, with two substrate-recognition domains, exploit the proximity effect to position specific subunits for productive pairing into assembly-competent intermediates. Second, CAPA chaperones display high substrate specificity, acting on only one or a narrow subset of adaptor subunits. This contrasts with folding chaperones such as Hsp70, which act broadly by binding generic hydrophobic regions[57]; CAPA chaperones instead use sequence-specific recognition to achieve subunit-level selectivity. Third, CAPA chaperones possess multiple substrate-binding modes, some of which mimic canonical cargo-binding motifs to engage adaptor subunits. The conserved μ1-binding region (YQPL) of MEA1 corresponds to a canonical tyrosine-based cargo motif (YXXΦ) on μ1/2[1,6]. Structural modeling indicates that MEA1 engages the tyrosine-based cargo-binding site on μ1/2 in a manner similar to native cargoes (Supplementary Fig. 14). Although AAGAB lacks canonical cargo motifs, it contains a sequence that inserts into the dileucine motif-binding pocket of AP1/2[26,30,58]. Likewise, CCDC32 harbors short linear motifs that resemble cargo motifs for AP2 binding[58]. By temporarily occupying these cargo-binding sites, CAPA chaperones block premature membrane cargo engagement and prevent promiscuous interactions with cytosolic proteins, ensuring that cargo recognition occurs only after full adaptor assembly. When these chaperones dissociate, the cargo-binding sites and other regulatory interfaces on APs become accessible, allowing the adaptor to engage cargo and downstream regulators such as adaptin ear-binding clathrin-associated protein (NECAP) and Fer/CIP4 Homology domain only (FCHo)[18,44].

While no assembly chaperones have yet been identified for AP3 or AP5, their conserved tetrameric architecture and subunit interdependence strongly suggest that their formation also requires dedicated assembly chaperones. Given that each of the four subunits of AP1

and AP2 requires stabilization by assembly chaperones, it is likely that AP3 and AP5 also depend on two or more assembly chaperones. Systematic identification of these chaperones will be essential to fully define the molecular logic of the CAPA pathway and to understand how its disruption contributes to human disease. More broadly, our findings raise the possibility that a substantial fraction of the proteome is devoted to encoding chaperones that guide the assembly of specific multisubunit protein complexes. Although individual complexes rely on distinct assembly chaperones, the dual-chaperone collision mechanism described here may represent a common principle for multisubunit complex assembly.

## Methods
### Cell culture
HEK 293 T cells, HeLa cells, and RPE1 cells were cultured in Dulbecco's Modified Eagle Medium (DMEM) supplemented with 10% Fetal Bovine Serum (FBS, Sigma, #F0926) and penicillin/streptomycin (Corning, #130-002-CI). All cell lines were maintained in a humidified 37 °C incubator with 5% $CO_2$.

### Plasmids
Mammalian expression plasmids encoding 3 × FLAG-tagged AAGAB and HA-tagged AP1/2 subunits were developed in our previous work[29,30]. To express 3 × FLAG-tagged human MEA1, a DNA fragment encoding human MEA1 and a C-terminal 3 × FLAG tag was subcloned into the NheI and SalI sites of the SHC003BSD-DelGFP vector (Addgene, #133301). To express ALFA-tagged MEA1, a DNA fragment encoding human MEA1 and a C-terminal ALFA tag was subcloned into the NheI and SalI sites of the SHC003BSD-DelGFP vector.

Plasmids expressing $His_6$-SUMO-tagged AAGAB, GST-tagged γ trunk domain (a.a. 1–595), and untagged FL σ1 were made in our previous work[28,30]. To express MEA1 in *E. coli*, Human *MEA1* was subcloned into the BamHI and XhoI sites of the pGEX-4T-3 vector (GE Healthcare) and the pRSF-Duet-SUMO expression vector[59,60]. *Ap1m1*, which encodes FL mouse μ1, was subcloned into the BamHI and XhoI sites of the pGEX-4T-3 vector or the NcoI and HindIII sites of pACYCDuet-1 vector. A DNA fragment encoding the trunk domain of human β1 (a.a. 1–584) was subcloned into the BamHI and XhoI sites of the pGEX-4T-3 vector. A GST-free version of this β1 plasmid was generated by removing the GST-encoding sequence using site-directed mutagenesis.

### Gene KO using CRISPR-Cas9
A gene was simultaneously targeted by two separate guide RNAs (gRNAs). Oligonucleotides of the two gRNAs were separately subcloned into the pLenti-CRISPR-V2 vector (Addgene, #52961) and the pLentiGuide-Hygro vector[30]. The CRISPR plasmids were transfected into HEK 293 T cells together with pAdVAntage (Promega, #E1711), pCMV-VSV-G (Addgene, #8454), and psPAX2 (Addgene, #12260) using a previously established procedure[30]. HEK 293 T cell culture media containing lentiviral particles were harvested daily for four days and centrifuged at 112,400 × *g* for 1.5 h in a SW28 rotor (Beckman Coulter). Viral pellets were resuspended in PBS and used to infect target cells. After lentiviral infection, cells were sequentially selected using 1 μg/mL puromycin (Sigma-Aldrich, #P8833) and 500 μg/mL hygromycin B (Thermo Fisher Scientific, #10687010). Oligonucleotide sequences of gRNAs targeting the human *MEA1* gene are 5′-GTTCTAGGAGGAGA-CACCAT-3′ and 5′-TACTGTTGCCATCCGGGCAG-3′. Sequences of gRNAs targeting the human *AP1G1* gene are 5′-TGCCAGCCCCCATCA-GATTG-3′ and 5′- ATTTTGCCACATTCCGACAT-3′.

### Flow cytometry
Cells were washed with KRH buffer (12 mM HEPES, pH 7.0, 121 mM NaCl, 4.9 mM KCl, 1.2 mM $MgSO_4$, and 0.33 mM $CaCl_2$) and blocked with KRH buffer containing 5% FBS at 4 °C. To measure surface FOLR1,

unpermeabilized cells were stained using PE-conjugated anti-FOLR1 antibodies (BioLegend, #908304, RRID: AB_2629795). After dissociation of cells from plates using Accutase, PE fluorescence was measured on a CyAn ADP Analyzer (Beckman Coulter). Data from populations of about 5,000 cells were analyzed using FlowJo, v. 10 (FlowJo, LLC) based on experiments run in biological triplicates.

### Immunostaining and imaging

Cells grown on coverslips were fixed using 4% paraformaldehyde (PFA) and permeabilized in PBS containing 5% FBS and 0.2% saponin. The 3 × FLAG-tagged AAGAB and MEA1 were detected using anti-FLAG M2 antibodies (Sigma-Aldrich, #F1804, RRID: AB_262044) and Alexa Fluor 488-conjugated secondary antibodies (Thermo Fisher Scientific, #A11008, RRID: AB_142165). ALFA-tagged MEA1 was labeled using mScarlet-anti-ALFA nanobodies[27]. The γ subunit was labeled using anti-γ-adaptin antibodies (Sigma, A4200, RRID: AB_476720) and anti-mouse Alexa Fluor 568-conjugated secondary antibodies (Thermo Fisher Scientific, #A11004, RRID: AB_2534072). β1 was labeled using anti-β1-adaptin antibodies (ProteinTech, #16932-1-AP, RRID: AB_2274034) and anti-rabbit Alexa Fluor 568-conjugated secondary antibodies (Thermo Fisher Scientific, #A11011, RRID: AB_143157). The plasma membrane was stained with CF405-conjugated Concanavalin A (Biotium, #29074). Confocal images were captured using a 100× oil immersion objective on a Nikon A1 laser scanning confocal microscope. In SIM, cells were grown on coverslips, labeled as in confocal microscopy, and imaged using a 100× oil immersion objective on a Nikon A1 SIM microscope.

### Immunoblotting and immunoprecipitation

To prepare whole cell lysates for immunoblotting, cells grown in 24-well plates were lysed in a SDS protein sample buffer (80 mM Tris, pH 6.8, 2% SDS, 10% glycerol, 0.0006% Bromophenol blue, and 0.1 M DTT). The samples were resolved on 8% Bis-Tris SDS-PAGE, transferred to PVDF membranes, and probed using antibodies. Primary antibodies used in immunoblotting included polyclonal anti-MEA1 antibodies (Bethyl Laboratories, #A305-779A, RRID: AB_2891676), polyclonal anti-STING antibodies (Proteintech, #19851-1-AP, RRID: AB_10665370), monoclonal anti-pSTING antibodies (Ser366, Cell Signaling Technology #19781, RRID: AB_2737062), monoclonal anti-pTBK1 antibodies (Ser172, Cell Signaling Technology, #5483, RRID:AB_10693472), polyclonal anti-clathrin heavy chain (CHC) antibodies (Cell Signaling Technology, #2410, RRID: AB_2083156), polyclonal anti-γ antibodies (Bethyl Laboratories, #A304-771A, RRID: AB_2620966), polyclonal anti-β1-adaptin antibodies (ProteinTech, #16932-1-AP, RRID: AB_2274034), polyclonal anti-μ1 antibodies (MyBioSource, #MBS712215), polyclonal anti-σ1 antibodies (Bethyl Laboratories, #A305-396A, RRID: AB_2631787), monoclonal anti-α-adaptin antibodies (BD Biosciences, #610502, RRID: AB_397868), polyclonal anti-β2 antibodies (Bethyl Laboratories, #A304-719A, RRID: AB_2620914), monoclonal anti-μ2 antibodies (BD Biosciences, #611350), polyclonal anti-σ2 antibodies (Abcam, #ab128950, RRID: AB_11140842), monoclonal anti-δ antibodies (DSHB, #SA4), monoclonal anti-ε antibodies (BD Biosciences, #612018), monoclonal anti-ζ antibodies (Thermo Fisher Scientific, #66533-1-IG), and monoclonal anti-α-tubulin antibodies (DSHB, #12G10, RRID: AB_1210456). Secondary antibodies used in immunoblotting included horseradish peroxidase (HRP)-conjugated anti-rabbit antibodies (Sigma-Aldrich, #A6154, RRID: AB_258284), and HRP-conjugated anti-mouse antibodies (Sigma-Aldrich, #A6782, RRID: AB_258315). FLAG-tagged proteins were directly detected using HRP-conjugated anti-FLAG M2 antibodies (Sigma-Aldrich, #A8592, RRID: AB_439702). HA-tagged proteins were directly detected using HRP-conjugated anti-HA antibodies (Roche, #12013819001, RRID: AB_390917).

In IP experiments, cells were lysed in an IP buffer (25 mM HEPES, pH 7.4, 150 mM NaCl, 10 mM $Na_3PO_4$, 2.7 mM KCl, 0.5% CHAPS, 1 mM DTT, and a protease inhibitor cocktail). After centrifugation, proteins were immunoprecipitated from cell extracts using anti-FLAG M2 antibodies and protein A/G agarose beads (Santa Cruz Biotechnology, #SC-2003). Immunoprecipitated proteins were resolved on 8% Bis-Tris SDS-PAGE and detected using immunoblotting. Intensities of protein bands on immunoblots were quantified using Fiji.

### Cell fractionation and CCV isolation

Cells from confluent 10-cm dishes were washed with PBS and resuspended in hypotonic buffer (10 mM HEPES, pH 7.4, 250 mM sucrose, 10 mM potassium acetate, 1.5 mM $MgCl_2$, 1 mM DTT, and a protease inhibitor cocktail). The cells were lysed using a Type B Dounce homogenizer, and the salt concentration was adjusted to physiological levels by adding high-salt buffer (25 mM HEPES, pH 7.4, 400 mM KCl, and 10% glycerol). Nuclei and intact cells were removed by centrifugation at $1000 \times g$ for 5 min in a microcentrifuge. The resulting supernatant was further centrifuged at $260,000 \times g$ for 30 min in a SW60 Ti rotor (Beckman Coulter) to separate cytosolic (supernatant) and membrane (pellet) fractions.

CCVs were isolated using a procedure adapted from a previous report[23]. Cells grown on 10-cm dishes were either untreated or treated with 2.5 μM diABZI (Selleckchem, #S8796) for 2 h. Cells were washed with PBS and harvested into Buffer A (0.1 M MES, pH 6.5, 0.2 mM EGTA, and 0.5 mM $MgCl_2$). Cell lysis was performed by repeated pipetting ( > 25 times) using a 5 mL syringe fitted with a 22-gauge needle. Lysates were centrifuged at $4100 \times g$ for 30 min, and supernatants were transferred to fresh tubes. RNase A was added at 50 μg/mL, followed by incubation on ice for 30 min. Samples were then subjected to ultracentrifugation at $260,000 \times g$ for 30 min in a SW60 Ti rotor. Resulting pellets were resuspended in 300 μL Buffer A, mixed with an equal volume of Buffer B (12.5% [w/v] Ficoll and 12.5% [w/v] sucrose in Buffer A), and centrifuged at $41,061 \times g$ for 25 min. Supernatants were collected, diluted fourfold with Buffer A, and centrifuged at $165,000 \times g$ for 30 min to obtain the CCV-enriched fraction.

### Recombinant protein expression and pull-down assays

Recombinant proteins were expressed in BL21 (DE3) E. coli (Stratagene, #230132) and purified using established procedures[30,61]. When the $OD_{600}$ of E. coli culture in 2 × YT media reached 0.6, 1 mM isopropyl β-D-1-thiogalactopyranoside (IPTG, GoldBio, #I2481C100) was added to induce protein expression. After three hours of incubation at 37 °C, bacteria were harvested and lysed using sonication. After centrifugation, proteins were isolated using glutathione beads (Thermo Fisher Scientific, #PI6101) or nickel beads (Thermo Fisher Scientific, #PI-88222). Proteins were resolved on 8% Bis-Tris SDS-PAGE and detected using Coomassie blue staining or immunoblotting.

### Structural prediction and analysis

The structures of MEA1:AP1 complexes were predicted using Alpha-Fold3 with default settings[62,63]. Five independent structural models were generated for each protein complex, and the quality of the predicted models was assessed through their ipTM scores, predicted template modeling (pTM) scores, predicted alignment error (PAE) plots, and predicted local distance difference test (pLDDT) scores[62,63]. A structural model for a protein complex was selected based on these scores and, whenever possible, its correlation with experimental data. PAE plots were generated using the PAE Viewer[50]. Structural analysis was conducted using UCSF ChimeraX (v1.7)[50] and PDBePISA[64].

### Quantification and statistics

Band intensities of immunoblots were quantified using ImageJ (v2.14) and normalized to α-tubulin bands. Data normalization was performed by setting the mean value of WT or KO data points as 100% or 1, and all data points were normalized to that mean value. Additional details are provided in figure legends. Student's t-test was used for comparisons

between two groups, while ANOVA was applied for analyses involving more than two groups, using GraphPad Prism (v9.0). Error bars indicate SD.

## Reporting summary

Further information on research design is available in the Nature Portfolio Reporting Summary linked to this article.

## Data availability

All data supporting the findings of this study are included in the main article and the Supplementary Information. Source data are provided with this paper.

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

## Acknowledgements

We thank Drs. James Hurley, Juan Bonifacino, Haoxi Wu, and Soyeon Park for reagents or advice. We thank James Orth for assistance with light microscopy. This work was supported by National Institutes of Health grants GM126960 (J.S.), DK142287 (J.S.) and GM138685 (Q.Y.). Publication of this article was partially funded by the University of Colorado Boulder Libraries Open Access Fund.

## Author contributions

C.W. and J.S. conceived the project. C.W., J.W., Y.O., and H.P. performed the experiments. C.W., J.W., Y.T., S.L., Q.Y., and J.S. analyzed the data. C.W., J.W., and J.S. wrote the manuscript with input from all authors.

## Competing interests

The authors declare no competing interests.
