## [Transparent Peer Review file · Nature Communications]

Regulation of AP1 adaptor assembly by the bi-handed chaperone MEA1

Corresponding Author: Professor Jingshi Shen

Version 0:

Reviewer comments:

Reviewer #1

(Remarks to the Author)

Using a combination of bioinformatic, genetic and biochemical approaches, Wan et al identify and characterize MEA1 as a novel assembly chaperone of AP1. MEA1 specifically associates with the μ 1 and β 1 subunits of AP1 promoting their assembly with gamma1 and sigma1 subunits which are stabilized by AAGAB, another assembly chaperone of AP1. This is an interesting and important manuscript. It is logically structured and clearly written. The experiments are well controlled and technically sound. The conclusions are supported by the results obtained. I have few comments the authors may want to consider when preparing the revised version.

1. The interplay between AAGAB and MEA1 should be addressed in more detail. In the cells lacking MEA1, all four AP1 subunits are strongly reduced - is this because levels of AAGAB are also reduced in these cells? Are AAGAB and MEA1 co-regulated?
2. Fig 1f shows coimmunoprecipitations among endogenous β 1, μ 1 and MEA1 proteins. First, it would be important to state the % of material shown in WCL compared to IP fractions, to be able to estimate the relative amounts of β 1 and μ 1 subunits bound to MEA1 at steady state. Second, it would be important to show western blots of gamma1 and sigma1 subunits in the same IPs, in order to judge the specificities of interactions.
3. The AlphaFold predicted interaction surfaces of MEA1 with β 1 and μ 1 subunits were only evaluated in in vitro pull down experiments (Figs S8 and S9). Their effects on MEA1 function should also be analysed in vivo, using assays described in Fig 2 when MEA1 KO cells were analysed.

Minor comment

It would be helpful for nonspecialists if acronym CCV (clathrin-coated vesicle?) was spelled out when used for the first time in the text

Reviewer #2

(Remarks to the Author)

This manuscript from Wan, et al. describes the discovery of Mea1 as a novel assembly chaperone for the AP1 clathrin adaptor complex. AP1 and the related AP2 clathrin adaptor complexes are important components of the clathrin coat, serving to direct cargo into transport vesicles. Using a targeted in silico AlphaFold screen, they identified Mea1 as a novel binder of the AP1 μ 1 and β 1 subunits. Using cell based assays and in vitro binding studies, they propose that Mea1 is an assembly chaperone for AP1. This model fits nicely with the role of another assembly chaperone, AAGAB, which stabilizes the gamma and sigma1 subunits of AP1. Therefore, this study represents the "missing piece" of AP1 assembly, nicely explaining how all 4 subunits are stabilized and assembled into the final heterotetramer. The labs of Qian Yin and Jingshi Shen have established this field, discovering the assembly role for AAGAB, CCDC32, and now Mea1. Overall, this manuscript is well written and easy to follow and the data overall supports the conclusions. I believe it is well suited for publication.

Here are my comments, which pertain primarily to the discussion of the predicted AlphaFold models and how they relate to the known structural biology of AP1.

1. The predicted structures could be better displayed in the main figures. What is missing right now is the context for where these binding sites are in the full AP1 complex.

2. Related to the above, there should be a figure with an overlay of AP1 mu1 bound to a tyrosine cargo, showing that Mea1 binds in the Y-cargo binding site.

3. I don't agree with the interpretation that Mea1 acts by shielding hydrophobic pockets on beta1 and mu1. First, the "pocket" on mu1 is simply the tyrosine cargo binding site, which is not particularly hydrophobic. This domain can be easily purified in *E. coli* and is highly stable on its own, so why would binding to the tyrosine cargo binding site help stabilize mu1? Next, the hydrophobic grooves they show for beta1 are highly misleading, and one is artifactual. The groove that binds to Mea1 AA 135-147 is not at an interface with gamma. It is a solvent-exposed groove that is accessible in both the closed (1W63.pdb) and open (4HMY.pdb) AP1 conformations. You can easily see this by doing an alignment using beta1 residues 400-550.

The other groove that binds to Mea1 AA 74-90 is an artifact from using the AlphaFold prediction with only beta1 AA 293-584. In this AF model, the C-terminus of beta1 (AA 542-584) folds back and makes a new groove by packing against the beta1 HEAT repeats. This "groove" doesn't exist in their trimer AF model, as beta1 residues are correctly packed against the beta1 N-terminus.

I think it's probably fine to suggest this model, but they should not use the beta1 AA 293-584 + Mea1 structure to make the figure to highlight the hydrophobic patch, and they should soften the language to indicate this is speculative.

4. Based on the above point, I think it is inappropriate to use the beta 293-584 sequence for an AlphaFold prediction. The authors should use a larger fragment of beta that correctly places the C-terminal tail, delete the tail, or simply use 1-584. We don't know the structure of beta1 in isolation, so an AF output that has a beta1 conformation significantly different from all known structures is misleading, without more validation and extensive discussion of the observed differences in the text.

5. A figure showing how Mea1 likely occupies the gamma binding site on beta1 would be helpful. Some image showing beta/gamma and the clash with Mea1 would highlight to the reader the likely mechanism for displacement upon gamma binding.

6. Mea1 AA 74-90 binds in the exact same binding spot that NECAP and Fcho use to bind to AP-2 (see 7OG1.pdb for Fcho2. For NECAP, density for its Ex-Tail domain was observed in EMD-20220 but not modeled due to low resolution). NECAP also binds to AP1, and therefore likely uses the same binding groove. This should be highlighted in the text and likely warrants a supplementary figure with an overlay of these regions.

7. The supplemental Excel spreadsheet is missing the vast majority of values for the predictions. This is likely an error in file upload, so this should be corrected to show the AF3 values for all predictions done in the in silico screen.

Congratulations on the interesting story!

-Rick Baker

UNC Chapel Hill

Reviewer #3

(Remarks to the Author)

The manuscript by Chun Wan and co-workers reports on the function of Male-Enhanced Antigen 1 (MEA1) as a bi-handed chaperone for the assembly of the heterotetrameric adaptor complex AP-1. MEA1 was identified as an interactor of AP-1 via a combination of proteomics screening and in silico interaction analysis. Loss of MEA1 resulted in altered trafficking of constitutive and induced cargoes of AP-1. Biochemical and in vitro tests as well as modeling further showed MEA1 to stabilize the mu and beta subunits of AP-1 and that MEA1 acted complementary to the previously identified AAGAB bi-handed chaperone, which stabilizes the other two AP-1 subunits. MEA1 and AAGAB stabilized hemicomplexes of AP-1 release their chaperones upon tetramerization. The authors present this example of an ATP-independent dual chaperone collision system as a general principle in the biogenesis of multiprotein complexes.

I liked reading the paper and I find that most experiments are adequately done and properly presented. Most of the experiments sufficiently support the conclusions drawn. I have some comments on the current version of this manuscript, which will help the authors to further increase the quality of their work and to increase the accessibility of their manuscript to a broad readership. I have outlined my remarks below.

Major remarks

Although the main point of the story concerns the role of assembly chaperones, the introduction does not adequately summarize the state-of-the art on this topic. Only in the last paragraph, AAGAB is mentioned, but without references despite this concerns very recent work from the same group. Recent work in *Arabidopsis* also revealed a role for the AAGAB homolog P34 in AP complex stabilization, showing broad conservation, which could be emphasized here. I would propose the authors to adapt the introduction section, focusing more on the key message of their story. I also found that the latest story of the Robinson lab, where they provide evidence for a retrograde trafficking role of AP-1 could be incorporated better, either in the introduction or in the discussion.

The choice to focus on MEA1 could be emphasized more. Now it is merely described as a strong candidate, but for the non-experienced reader, it would be better to indicate the specific parameters that allowed this candidate to be chosen (highest ipTM score for example).

The scheme depicted in Figure 2a does not depict FOLR1 nor its increased surface levels upon impairment of AP-1. This panel could be changed to better clarify the purpose of the experiments.

Quantifications throughout the manuscript are badly defined. For example, it is unclear, and not mentioned in the methods

unless I overlooked, how the quantifications in figure 2e and 2g are done. I would expect the values to be primarily normalized against those of tubulin as loading control. A dedicated paragraph on all the quantifications in methods and a clear description of how the quantifications were done in the figure legends would be advisable.

I do not really agree with the statement that localizing an individual AP complex subunit reflects the abundance of the full complex. Single subunits with membrane binding capacity will show similar localization as the full complex. This has been shown for plant AP-2 subunits. Also, hemicomplexes have been shown to exist and localize in worm. This statement therefore should be rephrased.

The statement that MEA1 only binds mu and beta and that it does not associate with the full adaptor complex is based on the data shown in figure 4 and absence of identification from proteomics data. These experiments do not suffice to make such a strong statement, please adapt or provide additional data.

The data on the structural models of MEA1:AP-1 in figure 5 would optimally be compared to a control as this would allow better evaluation of what is shown. Performing the modeling with MEA1:AP-2 would be a proper control here.

A weak point in the paper concerns the interaction between MEA1 and the AP-2 mu and beta subunits, given that this seems hardly linked to a stabilizing function for those subunits. The interaction between MEA1 and AP-2 was shown using HA-tagged subunits. I could not find data in this manuscript comparing the expression levels of the HA-tagged AP-2 subunits compared to the endogenous ones and I wonder why these HA-tagged subunits were used, given that antibodies against the endogenous subunits are available. It would help if the authors could show, or emphasize the evidence that MEA1 interacts with the endogenous AP-2 subunits. Moreover, there is quite some data in the manuscript showing that specific mutations in MEA1 disrupt binding to AP-1 subunits. Is that similar for AP-2 and how is the binding interface (shown in figure 5b and 5d for AP-1) between MEA1 and AP-2? I do believe it to be important to include data although we do not fully comprehend their relevance at this point. In this particular case however, some additional independent confirmation of the MEA1:AP-2 interaction would be advisable.

Minor remarks

I would spell out MEA1 in the abstract upon first use. Similarly, STING and AAGAB should also be spelled out if word counts allow for this.

In figure 1, the Western showing the input is placed after the Western showing the IP fractions. I would reverse the order here as this is more intuitively.

Figure S1: I am missing the data on the sequences that were used to generate the alignment.

The statement that MEA1 is essential for trafficking of AP-1 cargoes should be tuned down as there is still some pSTING in the CCV fraction, which is higher than that present in the AP-1 KO.

The statement (Given the similar KO phenotypes of MEA1 and AAGAB... remaining subunits) is oddly positioned in the text as the specificity of both chaperones is only determined later.

In figure 7, it is unclear to me what is meant with the statement that the hemicomplexes are not depicted. They are depicted but not according to the structural model?

In the discussion, it is mentioned that the binding of MEA1 corresponds to a canonical AP-1 cargo motif. This statement is odd as I do not recall a difference between an AP-1 and an AP-2 tyrosine binding motifs. Similarly, a few lines below, it is stated that AAGAB contains a sequence that inserts into the dileucine motif of AP-2. Is this then similar for AP-1? These sentences could be rephrased as they are now hinting towards adaptor complex specificity of these chaperones, which is not the case.

Version 1:

Reviewer comments:

Reviewer #1

(Remarks to the Author)

In the revised version, the authors addressed the comments raised by the reviewers in a satisfactory manner. It is a pity that the antibodies against endogenous MEA1, AP1-β1 and AP1-μ1 are either unable to immunodeplete their respective antigens from the whole cell lysates or the amounts of the antibodies used in the experiment were insufficient for immunodepletion – it would have been nice to be able judge the amount of the assembly intermediates compared to the fully assembled complex at the steady state. Also, I may be missing something but it is unclear to me how newly modified Fig 2a illustrates aberrant accumulation of FOLR1 on the cell surface in the absence of AP1.

Reviewer #2

(Remarks to the Author)

The reviewers have addressed all of my concerns. The manuscript is ready for acceptance.

-Rick Baker
UNC Chapel Hill

Reviewer #3

(Remarks to the Author)

The authors have adequately answered to my previously raised concerns. I have no further comments. Many congratulations on an very nice paper.

Point-to-point Responses to Reviewers' Comments

Reviewer 1

“1. The interplay between AAGAB and MEA1 should be addressed in more detail. In the cells lacking MEA1, all four AP1 subunits are strongly reduced - is this because levels of AAGAB are also reduced in these cells? Are AAGAB and MEA1 co-regulated?”

Response: This is an excellent point. Based on the Reviewer's suggestion, we examined AAGAB levels in *MEA1* KO cells and observed no change. Likewise, MEA1 levels remained unchanged in *AAGAB* KO cells. These data indicate that the two chaperones are not dependent on each other for their stability. These new results are shown in Figure S2c-d of the revised manuscript. We also revised the text (page 13, paragraph 2) to discuss these findings. We thank the Reviewer for this important suggestion.

“2. Fig 1f shows coimmunoprecipitations among endogenous $\beta 1$, $\mu 1$ and MEA1 proteins. First, it would be important to state the % of material shown in WCL compared to IP fractions, to be able to estimate the relative amounts of $\beta 1$ and $\mu 1$ subunits bound to MEA1 at steady state. Second, it would be important to show western blots of $\gamma 1$ and $\sigma 1$ subunits in the same IPs, in order to judge the specificities of interactions.”

Response: Based on the Reviewer's advice, we revised Figure 1 to include the percentage of input materials and added $\sigma 1$ and $\gamma 1$ to the same IPs.

“3. The AlphaFold predicted interaction surfaces of MEA1 with $\beta 1$ and $\mu 1$ subunits were only evaluated in in vitro pull down experiments (Figs S8 and S9). Their effects on MEA1 function should also be analysed in vivo, using assays described in Fig 2 when MEA1 KO cells were analysed.”

Response: In response to the Reviewer's suggestion, we tested how these mutations affect MEA1 function in cells. We observed that, when expressed at levels comparable to WT MEA1, the mutants failed to restore AP1 or proper STING signaling in *MEA1* KO cells. As shown in Figure S11 of the revised manuscript, these new results are fully consistent with our structural models and biochemical data. We also revised the text (page 12, paragraph 2) to discuss these findings.

“Minor comment: It would be helpful for nonspecialists if acronym CCV (clathrin-coated vesicle?) was spelled out when used for the first time in the text.”

Response: Following the Reviewer's recommendation, we spelled out CCV at its first occurrence.

Finally, we thank the Reviewer for these excellent suggestions and for supporting the publication of this work.

Reviewer 2

“1. The predicted structures could be better displayed in the main figures. What is missing right now is the context for where these binding sites are in the full AP1 complex.”
“2. Related to the above, there should be a figure with an overlay of AP1 μ 1 bound to a tyrosine cargo, showing that Mea1 binds in the Y-cargo binding site.”

Response: Here we address these two points together because they are conceptually linked. Based on the Reviewer’s suggestions, we added two additional figures (Figures S13 and S14) to show the relevant structural comparisons. In Figure S13, we compared the MEA1:AP1 structural models with crystal structures of the full AP1 adaptor. One conclusion from these comparisons is that MEA1 occupies the γ -binding site on β 1, indicating that MEA1 must be released before γ can associate with β 1 to form the full AP1 adaptor. This conclusion is consistent with our biochemical and imaging data. Another conclusion from Figure S14 is that a region in MEA1 mimics a tyrosine cargo motif when engaging μ 1/2. We also revised the text to discuss the implications of these findings (page 14, paragraph 1; page 17, paragraph 2). We thank the Reviewer for these excellent suggestions.

“3. I don’t agree with the interpretation that Mea1 acts by shielding hydrophobic pockets on beta1 and mu1. First, the “pocket” on mu1 is simply the tyrosine cargo binding site, which is not particularly hydrophobic. This domain can be easily purified in E. coli and is highly stable on its own, so why would binding to the tyrosine cargo binding site help stabilize mu1? Next, the hydrophobic grooves they show for beta1 are highly misleading, and one is artifactual. The groove that binds to Mea1 AA 135-147 is not at an interface with gamma. It is a solvent-exposed groove that is accessible in both the closed (1W63.pdb) and open (4HMY.pdb) AP1 conformations. You can easily see this by doing an alignment using beta1 residues 400-550. The other groove that binds to Mea1 AA 74-90 is an artifact from using the AlphaFold prediction with only beta1 AA 293-584. In this AF model, the C-terminus of beta1 (AA 542-584) folds back and makes a new groove by packing against the beta1 HEAT repeats. This “groove” doesn’t exist in their trimer AF model, as beta1 residues are correctly packed against the beta1 N-terminus. I think it’s probably fine to suggest this model, but they should not use the beta1 AA 293-584 + Mea1 structure to make the figure to highlight the hydrophobic patch, and they should soften the language to indicate this is speculative.”

“4. Based on the above point, I think it is inappropriate to use the beta 293-584 sequence for an AlphaFold prediction. The authors should use a larger fragment of beta that correctly places the C-terminal tail, delete the tail, or simply use 1-584. We don’t know the structure of beta1 in isolation, so an AF output that has a beta1 conformation significantly different from all known structures is misleading, without more validation and extensive discussion of the observed differences in the text.”

Response: Here we address these two points together because they are conceptually linked. We fully agree with the Reviewer on both points. The Reviewer correctly pointed out that MEA1 binding to the μ 1 cargo-binding site does not constitute shielding; rather, this interaction is

highlighted because it represents a characteristic engagement mode used by AP assembly chaperones. We revised Figure S5c to better illustrate that the shielding effect pertains to a different region. Based on the Reviewer's suggestion, we revised Figures 5 and S5 to use the entire core domain (a.a. 1–584) of $\beta 1$ for structural prediction. These structural models suggest that MEA1 may shield hydrophobic regions on $\beta 1$. While these observations are consistent with known functions of assembly chaperones, we fully agree with the Reviewer that this interpretation remains speculative. We have revised the text to explicitly discuss these points and softened the tone of the statements (page 12, paragraph 1).

“5. A figure showing how Mea1 likely occupies the gamma binding site on beta1 would be helpful. Some image showing beta/gamma and the clash with Mea1 would highlight to the reader the likely mechanism for displacement upon gamma binding.”

Response: Based on the Reviewer's suggestion, we added a new figure (Figure S13) to show the structural model that MEA1 occupies γ -binding site on $\beta 1$, indicating that MEA1 must be released before γ can associate with $\beta 1$ to form the full AP1 adaptor. This conclusion is fully consistent with our biochemical and imaging data. We also revised the text to discuss these new data (page 14, paragraph 1). We thank the Reviewer for this excellent advice.

“6. Mea1 AA 74-90 binds in the exact same binding spot that NECAP and Fcho use to bind to AP-2 (see 7OG1.pdb for Fcho2. For NECAP, density for its Ex-Tail domain was observed in EMD-20220 but not modeled due to low resolution). NECAP also binds to AP1, and therefore likely uses the same binding groove. This should be highlighted in the text and likely warrants a supplementary figure with an overlay of these regions.”

Response: The Reviewer raised an intriguing point. The overlap of MEA1, NECAP, and FCHO binding sites on AP1/2 is consistent with our model that MEA1 dissociation is a prerequisite for full AP1/2 formation and for subsequent engagement of downstream factors in clathrin-mediated trafficking. Although detailed analyses of these downstream factors are beyond the scope of this study, they represent an important direction for future research. We revised the Discussion section (pages 17-18, paragraph 2) to explicitly discuss this point.

“7. The supplemental Excel spreadsheet is missing the vast majority of values for the predictions. This is likely an error in file upload, so this should be corrected the show the AF3 values for all predictions done in the in silico screen.”

Response: We have updated the Excel spreadsheet to include AF3 values for all candidates that met the initial filtering criteria.

Finally, we thank the Reviewer again for sharing his/her deep expertise in AP1 biology and membrane trafficking. The manuscript has been significantly strengthened as a result of these revisions.

Reviewer 3

“Although the main point of the story concerns the role of assembly chaperones, the introduction does not adequately summarize the state-of-the art on this topic. Only in the last paragraph, AAGAB is mentioned, but without references despite this concerns very recent work from the same group. Recent work in Arabidopsis also revealed a role for the AAGAB homolog P34 in AP complex stabilization, showing broad conservation, which could be emphasized here. I would propose the authors to adapt the introduction section, focusing more on the key message of their story. I also found that the latest story of the Robinson lab, where they provide evidence for a retrograde trafficking role of AP-1 could be incorporated better, either in the introduction or in the discussion.”

Response: These are excellent suggestions. Based on the Reviewer’s advice, we revised the Introduction to incorporate the function and mechanism of AAGAB in AP assembly (page 4, paragraph 1). We also cited the plant p34/AAGAB paper in the Introduction and added the Robinson JCB paper to the beginning of the Introduction section, which nicely demonstrates a retrograde trafficking role for AP1 in mammalian cells (page 3, paragraph 1). With these changes, the logic and flow of the manuscript have been substantially improved, and we thank the Reviewer for these suggestions.

“The choice to focus on MEA1 could be emphasized more. Now it is merely described as a strong candidate, but for the non-experienced reader, it would be better to indicate the specific parameters that allowed this candidate to be chosen (highest ipTM score for example).”

Response: We fully agree with the Reviewer that the rationale for prioritizing MEA1 should be made more explicit. In the revised manuscript, we clarified that MEA1 was selected based on multiple convergent criteria rather than a single parameter. MEA1 showed high ipTM scores in AlphaFold-based predictions, appeared in both the μ 1 and β 1 interactomes, and satisfied all of our initial filtering criteria. The combination of these orthogonal features distinguished MEA1 from other candidates and motivated us to focus on it in this study. We revised the text accordingly (page 6, paragraph 2) to discuss these points. We thank the Reviewer for prompting this clarification.

“The scheme depicted in Figure 2a does not depict FOLR1 nor its increased surface levels upon impairment of AP-1. This panel could be changed to better clarify the purpose of the experiments.”

Response: In response to the Reviewer’s advice, we revised Figure 2a and its legend to include FOLR1 in the diagram and to discuss the consequences of disrupting AP1 function.

“Quantifications throughout the manuscript are badly defined. For example, it is unclear, and not mentioned in the methods unless I overlooked, how the quantifications in figure 2e and 2g are done. I would expect the values to be primarily normalized against those of tubulin as loading

control. A dedicated paragraph on all the quantifications in methods and a clear description of how the quantifications were done in the figure legends would be advisable.”

Response: Based on the Reviewer’s suggestion, we added a dedicated section on data quantification to the Methods section. We also revised figure legends to clarify how the data were analyzed, normalized, and presented. The Reviewer was correct that protein band intensities from immunoblots were normalized to α -tubulin. Moreover, all raw data are provided in the Supplementary Datasets. We thank the Reviewer for this important suggestion.

“I do not really agree with the statement that localizing an individual AP complex subunit reflects the abundance of the full complex. Single subunits with membrane binding capacity will show similar localization as the full complex. This has been shown for plant AP-2 subunits. Also, hemicomplexes have been shown to exist and localize in worm. This statement therefore should be rephrased.”

Response: The Reviewer correctly pointed out that AP2 subcomplexes exist in plants and worms. While such subcomplexes have not been unambiguously demonstrated for AP1, we fully agree that our original wording could be rephrased to avoid confusion. Therefore, we revised the text to soften the tone and more accurately reflect the literature and our own findings (page 14, paragraph 1).

“The statement that MEA1 only binds mu and beta and that it does not associate with the full adaptor complex is based on the data shown in figure 4 and absence of identification from proteomics data. These experiments do not suffice to make such a strong statement, please adapt or provide additional data.”

Response: While our conclusion that MEA1 acts prior to full AP1 formation is supported by multiple lines of evidence, we fully agree with the Reviewer that we should avoid an overly strong statement. Accordingly, we revised the text to soften the tone and more accurately reflect our findings (page 10, paragraph 1).

“The data on the structural models of MEA1:AP-1 in figure 5 would optimally be compared to a control as this would allow better evaluation of what is shown. Performing the modeling with MEA1:AP-2 would be a proper control here.”

Response: This is an excellent point. Based on the Reviewer’s advice, we added a new figure (Fig. S6) showing structural models of the MEA1:AP2 complex and its comparison with those of MEA1:AP1. By contrast, AlphaFold did not predict high-confidence structural models between MEA1 and other APs such as AP3, consistent with our genetic data indicating that MEA1 does not regulate these APs.

“A weak point in the paper concerns the interaction between MEA1 and the AP-2 mu and beta

subunits, given that this seems hardly linked to a stabilizing function for those subunits. The interaction between MEA1 and AP-2 was shown using HA-tagged subunits. I could not find data in this manuscript comparing the expression levels of the HA-tagged AP-2 subunits compared to the endogenous ones and I wonder why these HA-tagged subunits were used, given that antibodies against the endogenous subunits are available. It would help if the authors could show, or emphasize the evidence that MEA1 interacts with the endogenous AP-2 subunits. Moreover, there is quite some data in the manuscript showing that specific mutations in MEA1 disrupt binding to AP-1 subunits. Is that similar for AP-2 and how is the binding interface (shown in figure 5b and 5d for AP-1) between MEA1 and AP-2? I do believe it to be important to include data although we do not fully comprehend their relevance at this point. In this particular case however, some additional independent confirmation of the MEA1:AP-2 interaction would be advisable.”

Response: While this manuscript primarily focuses on AP1, we fully agree with the Reviewer that AP2 should be further characterized. Based on the Reviewer’s suggestion, we used AlphaFold to predict the MEA1:AP2 complex and compared this structural model with that of MEA1:AP1. As shown in Figure S6, these structural models indicate that MEA1 binds to AP1 and AP2 in a similar manner. To further address the Reviewer’s comment, we performed additional co-IP experiments to detect interactions between endogenous MEA1 and AP2. We observed that endogenous MEA1 selectively interacted with endogenous β 2 and μ 2, consistent with our co-IP data using tagged proteins. These new data are shown in Figure S3b. We also revised the Results section to discuss these findings (page 9, paragraph 2; page 11, paragraph 1).

“I would spell out MEA1 in the abstract upon first use. Similarly, STING and AAGAB should also be spelled out if word counts allow for this.”

Response: Based on the Reviewer’s suggestion, we spelled out these words at their first appearance in the abstract.

“In figure 1, the Western showing the input is placed after the Western showing the IP fractions. I would reverse the order here as this is more intuitively.”

Response: Following the Reviewer’s recommendation, we switched the order of the two panels, which indeed improved the clarity of the data. We thank the Reviewer for this important suggestion.

“Figure S1: I am missing the data on the sequences that were used to generate the alignment.”

Response: Based on the Reviewer’s suggestion, we added UniProt identifiers to the figure.

“The statement that MEA1 is essential for trafficking of AP-1 cargoes should be tuned down as there is still some pSTING in the CCV fraction, which is higher than that present in the AP-1 KO.”

Response: We fully agree with the Reviewer's point. We revised the text (page 9, paragraph 3) to soften the statement and more accurately reflect our results.

"The statement (Given the similar KO phenotypes of MEA1 and AAGAB... remaining subunits) is oddly positioned in the text as the specificity of both chaperones is only determined later."

Response: In response to the Reviewer's comment, we revised the paragraph (page 10, paragraph 1) to improve the flow of the text.

"In figure 7, it is unclear to me what is meant with the statement that the hemicomplexes are not depicted. They are depicted but not according to the structural model?"

Response: The statement was intended to indicate that the diagram did not depict how the hemicomplexes are formed. Based on the Reviewer's comment, we removed this statement to avoid confusion.

"In the discussion, it is mentioned that the binding of MEA1 corresponds to a canonical AP-1 cargo motif. This statement is odd as I do not recall a difference between an AP-1 and an AP-2 tyrosine binding motifs. Similarly, a few lines below, it is stated that AAGAB contains a sequence that inserts into the dileucine motif of AP-2. Is this then similar for AP-1? These sentences could be rephrased as they are now hinting towards adaptor complex specificity of these chaperones, which is not the case."

Response: The Reviewer correctly pointed out that the cargo-recognizing sites of AP1 and AP2 are similar, and we did not intend to imply binding specificity in this context. Based on the Reviewer's suggestion, we revised these statements in the Discussion section (page 17, paragraph 2) to avoid confusion.

Finally, we thank the Reviewer for the excellent suggestions, which have substantially improved the manuscript.

Point-to-point Responses to Reviewers' Comments

All reviewers were satisfied with our revisions and did not request any additional changes to the manuscript.